# The genomic epidemiology of shigellosis in South Africa

George E. Stenhouse [1] ✉, Karen H. Keddy [2], Rebecca J. Bengtsson[1], Neil Hall[3], Anthony M. Smith [4,5], Juno Thomas [4], Miren Iturriza-Gómara[1] & Kate S. Baker [1,6] ✉

Shigellosis, a leading cause of diarrhoeal mortality and morbidity globally, predominantly affects children under five years of age living in low- and middle-income countries. While whole genome sequence analysis (WGSA) has been effectively used to further our understanding of shigellosis epidemiology, antimicrobial resistance, and transmission, it has been under-utilised in sub-Saharan Africa. In this study, we applied WGSA to large sub-sample of surveillance isolates from South Africa, collected from 2011 to 2015, focussing on *Shigella flexneri* 2a and *Shigella sonnei*. We find each serotype is epidemiologically distinct. The four identified *S. flexneri* 2a clusters having distinct geographical distributions, and antimicrobial resistance (AMR) and virulence profiles, while the four sub-Clades of *S. sonnei* varied in virulence plasmid retention. Our results support serotype specific lifestyles as a driver for epidemiological differences, show AMR is not required for epidemiological success in *S. flexneri*, and that the HIV epidemic may have promoted *Shigella* population expansion.

Shigellosis is the second leading cause of diarrhoeal death, globally[1,2], with the greatest disease burden on those living in low- and middle-income countries[1,2]. Of further concern is the high burden of disease among children under five years old (around 30% of cases and 38% of deaths), with repeat childhood infections bringing long-term health consequences, including stunted growth, impaired cognitive abilities, and chronic, functional bowel disorders[3–10].

The causative agents, a group of Gram-negative bacteria called *Shigella*, are convergently evolved specialised pathovars of *E. coli*[11]. Made up of four serogroups (*S. flexneri, S. sonnei, S. boydii* and *S. dysenteriae*), they share an enteroinvasive pathogenesis mediated by acquisition of the large invasion, or virulence, plasmid (pINV)[11]. Each serogroup is epidemiologically distinct, though *S. flexneri* and *S. sonnei* are of primary importance due to their global distribution and high prevalence (accounting for approximately 90% of the global shigellosis burden)[12–15]. The two serogroups are also genetically distinct; *S. sonnei* has only one serotype and is highly clonal, while *S. flexneri* is far more diverse genetically and serotypically (having >20 different serotypes)[16]. Whole genome sequence analysis (WGSA) on large temporo-geographical collections of both serogroups has established new genomic nomenclatures, complementing traditional serological distinctions. These differentiate *S. flexneri* into eight phylogroups, and *S. sonnei* into five Lineages, and subsequently Clades and sub-Clades[12,13,17].

The serogroups are also thought to have distinct ecologies, with a switch from *S. flexneri* dominance to *S. sonnei* being linked with increasing industrialisation[14]. This may be due to reduced *Plesiomonas shigelloides* exposure in industrialised settings leading to reduced cross-protection against *S. sonnei* from their shared O-antigen[18,19]. Though differences in pathogen lifestyle may also be involved, genetic

[1]Clinical Infection, Microbiology, and Immunology, University of Liverpool, Liverpool, UK. [2]Independent Consultant, Johannesburg, South Africa. [3]Earlham Institute, Norwich Research Park, NR4 7UZ Norwich, UK. [4]Centre for Enteric Diseases, National Institute for Communicable Diseases (NICD), Division of the National Health Laboratory Service (NHLS), Johannesburg, South Africa. [5]Department of Medical Microbiology, Faculty of Health Sciences, University of Pretoria, Pretoria, South Africa. [6]Department of Genetics, University of Cambridge, CB23EH Cambridge, UK. ✉e-mail: georgiesten@googlemail.com; kb827@cam.ac.uk

differences in *S. sonnei* point to additional adaptations towards becoming an obligate pathogen (reduced pINV stability and O-antigen capsule expression)[20,21]. Understanding the epidemiological dynamics and pathogen ecologies of these two important pathogens is critical to their control.

Multidrug resistance (MDR) is widespread in both *S. flexneri* and *S. sonnei* and has been important in driving their epidemiology[12,13,17,22]. Increasing antimicrobial resistance (AMR) is such a risk to ongoing effective treatment that fluoroquinolone-resistant strains are classed as a WHO priority AMR pathogen and *Shigella* a recommended priority target organism for vaccines for AMR[12,13,23–27]. Genomic analyses reveal the genetic basis for AMR and enables tracking of their dissemination through the population. Many resistance determinants are carried on mobile genetic elements, such as the *Shigella* resistance locus (SRL), an important MDR element encoding resistance against ampicillin, chloramphenicol, streptomycin and tetracycline[26,27], which is mobile alone or as part of the larger SRL pathogenicity island (PAI). The highest impact on number of deaths due to increased AMR is known to be in LMIC[28].

Despite the greatest shigellosis disease burden falling on those living in sub-Saharan Africa (SSA) and Asia, the use of WGSA for studying shigellosis in SSA has been under-utilised. The largest study, to date, using WGSA on African isolates was The Global Enteric Multicentre Study (GEMS), which confirmed *S. flexneri* and *S. sonnei* to be the first and second most prevalent serogroups in three of four study countries (The Gambia, Mozambique, Mali)[5,29,30]. Further WGSA of the GEMS study isolates showed multiple Phylogroups of *S. flexneri* in each participating African country, highlighting the diversity of circulating strains[16]. While this provides a useful snapshot into the diversity of *Shigella* strains circulating regionally, detailed national studies are needed to further unpick the epidemiology of shigellosis in the region.

Non-genomic studies from across SSA highlight the importance of endemic disease in the region and confirm *S. flexneri* is typically the dominant serogroup of *Shigella*[31–45]. There is variation in shigellosis incidence, sub-continentally and nationally, with national attributable disease burden estimates of up to 63.3% of diarrhoeal cases, and intra-national variation as high as 34%[46,47]. The available data show AMR *Shigella* are present across the entire sub-continent, and MDR, where reported, is more common in *S. flexneri*[31,46,48–50]. South Africa has a high incidence (2.7 hospitalisations/100,000 persons per year) of shigellosis, caused predominantly by *S. flexneri* and *S. sonnei*, discovered through routine, national shigellosis surveillance, enabling detailed examination of the national epidemiology of serotypes from both globally important serogroups within SSA[51–58].

In this study, we use WGSA to examine the epidemiology, ecology, and accessory genome dynamics of shigellosis in the sub-Saharan nation of South Africa using isolates collected as part of the national surveillance between 2011 and 2015. We focus on the two most prevalent serotypes (*S. flexneri* 2a and *S. sonnei*) that caused approximately 70% of shigellosis cases in the country during the study period[51–58] and find evidence for distinct ecologies and AMR dynamics between the serotypes.

## Results

### The comparative epidemiology of *S. sonnei* and *S. flexneri* 2a in South Africa

We used reported case numbers from the annual surveillance reports by the Group for Enteric, Respiratory and Meningeal Diseases Surveillance in South Africa (GERMS-SA) to characterise the epidemiology of shigellosis in the country. Cases were unevenly distributed across the country, relative to provincial population size. More cases than expected were reported in Gauteng (+13.9%) and Western Cape (+11.8%) ($\chi^2$(26, $n$ = 5078.01) = 2156, $p$ = 0.0000), with the converse in Limpopo (−9.8%), Mpumalanga (−5.2%), North West (−5.7%) and

KwaZulu–Natal (−5.6%)[54–56]. This may represent an imperfect surveillance system, differences in healthcare availability, differences in health-seeking behaviour, or genuine differences in shigellosis incidence across the country.

To conduct genomic epidemiological analysis, we created a representative surveillance subsample of the two most burdensome serotypes, *S. flexneri* 2a ($n$ = 286) and *S. sonnei* ($n$ = 275), collected between 2011 and 2015. Statistical analysis supported the representativeness of our sample relative to disease incidence (reported case numbers), with no significant difference between observed and expected isolate numbers by either province ($\chi^2$(26, $n$ = 353) = 19.02, $p$ = 0.890) or year (2011–2013, data unavailable 2014-2015; $\chi^2$(5, $n$ = 353) = 2.75, $p$ = 0.474) (Fig. 1A). When stratifying by serotype, however, there were slightly more *S. sonnei* than expected from Gauteng (+15%; $\chi^2$(8, $n$ = 253) = 35.69, $p$ = 0.0000) and *S. flexneri* 2a from the Western Cape (+11%; $\chi^2$(8, $n$ = 260) = 31.091, $p$ = 0.0001) (Fig. 1A). Similar over-representations were also observed in the reported cases; between 2011 and 2013, 44–66% of *S. sonnei* cases were from Gauteng and 22–38% of *S. flexneri* 2a were from the Western Cape[54–56]. Thus, this supports serotype-specific geographic distributions in the country, across the study period, rather than an over representation of these serotypes from these regions due to sub-sampling bias.

To look at the potential influence of urbanisation on serotype distribution, we categorised districts by level of urbanisation according to methods laid out by the European Commission (March 2020). Most study isolates were from urban districts (72% total; 69% *S. flexneri* 2a and 76% *S. sonnei*) and the fewest from mixed districts (9%; 8% *S. flexneri* 2a and 10% *S. sonnei*). Greater representation of rural districts than mixed could be due to socioeconomic factors influencing disease incidence or differential surveillance efforts (Fig. 2B, C). Consistent with reports of *S. sonnei* being a pathogen of urban and urbanising areas[14], statistical analysis showed *S. sonnei* was more likely to be from mixed and urban districts than *S. flexneri* 2a ($\chi^2$(2, $n$ = 513) = 7.489, $p$ = 0.0024), extending the granularity of this international observation to be relevant at a national level.

We also explored possible associations between available patient demographic characteristics and shigellosis caused by either serotype. The median age of infection (all isolates) was higher in females (5 years old, IQR = 2–30) than males (3.5 years old, IQR = 1.25–11.75; Mann–Whitney $U$ $W$ = 18685, $p$ = 0.0027) (raw data shown by serotype Fig. 1). This age difference was more pronounced in *S. sonnei* (3-year difference in median patient age) than *S. flexneri* 2a (1-year difference); reflected as a statistically significant difference being observed only in *S. sonnei* when stratifying by serotype (Mann–Whitney $U$ test *S. sonnei*: $W$ = 5739.5, $p$ = 0.0072; *S. flexneri*: $W$ = 7487, $p$ = 0.0789) (Fig. 1). Secondary infection within the home has been shown to a driver of *S. sonnei* transmission[59,60], and it may be that the higher age in females results from onward infection during caring responsibilities for ill children in this setting. We found no association between sex and serotype.

### *Shigella flexneri* 2a population dynamics

We used phylogenetics to examine and contextualise the South African *S. flexneri* 2a population structure. All *S. flexneri* 2a study isolates belonged to Phylogroup 3 (PG3) (Supplementary fig. 1), according to a single nucleotide polymorphism (SNP)-based maximum likelihood (ML) phylogeny of our isolates ($n$ = 260) and global reference isolates ($n$ = 116). For a more granular visualisation of the population structure, we generated a PG3 only ML phylogeny of our isolates and PG3 reference isolates ($n$ = 116) (Fig. 2A). We also used Bayesian phylogenetics to characterise the population dynamics of *S. flexneri* 2a in South Africa through time. We identified four co-existing strains, two of which were likely recent introductions, and found evidence of population diversification beginning in the 1990s (Fig. 2B, Supplementary fig. 3). The geographical clustering of the South African isolates in the

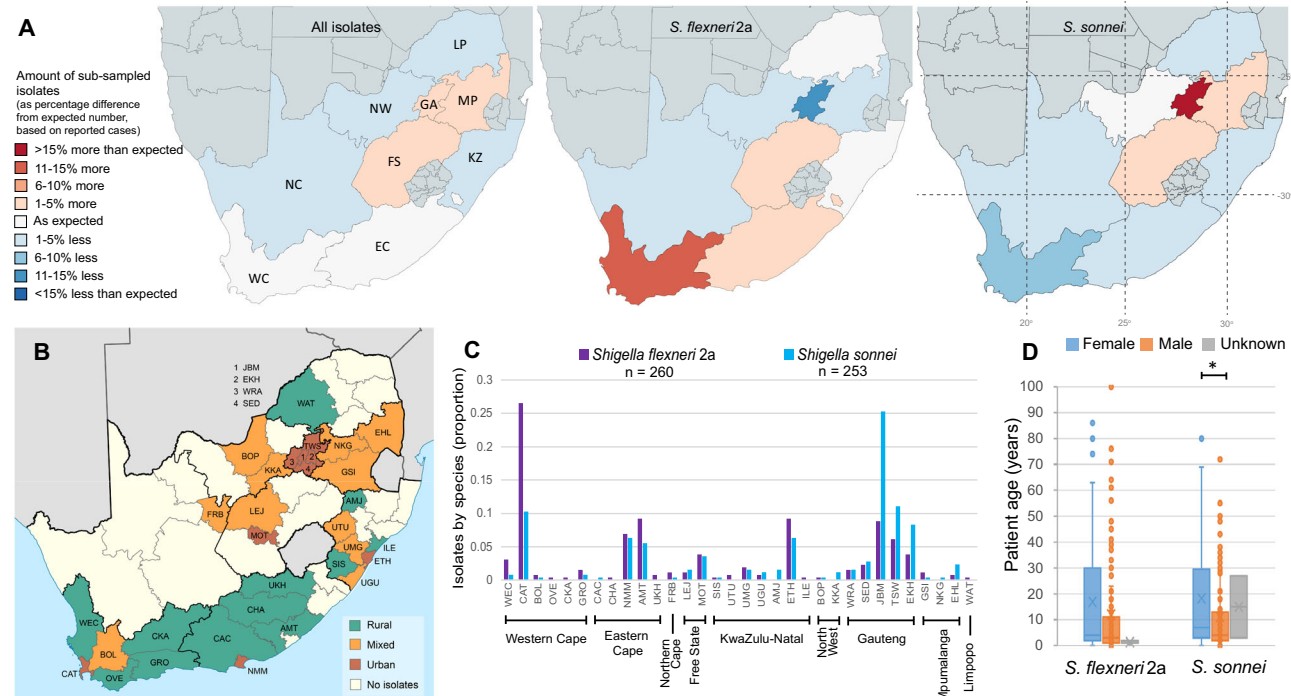

**Fig. 1 | Representativeness, and geographic and patient demographic features of the surveillance subsample. A** Provincial representation the surveillance cases within isolate subsample. The under or over representation of cases by sub-sampled isolates ($\frac{O-E}{E}$, where $O$ = actual number of sub-sampled isolates and $E$ = expected number of isolates based on reported case numbers) is shown for all isolates (left), *S. flexneri* 2a (middle), and *S. sonnei* (right), coloured by percentage divergence from expected number of isolates based on reported cases, according to the inlaid key. NB: data are shown for 2011 to 2013 due to a lack of reported data from 2014 and 2015. Latitude and longitude are overlaid on *S. sonnei* map. **B** Urbanisation of sampled districts, level of urbanisation indicated by colour as shown in inlaid key (bottom right). **C** Proportion of subsample by district (*x* axis) and province (overlying horizontal bars) of origin and serotype (*S. flexneri* 2a = purple, *S. sonnei* = blue). **D** Distribution of patient ages among the surveillance subsample by *Shigella* serotype (*x* axis) and sex (female = blue and male = orange, unknown sex = grey). Data are presented as Box plots: *S. flexneri* 2a (male and female, respectively) minima = 0 and 0, maxima = 86 and 100, median = 3 and 4, IQR (bounds of box) = 1–11 and 2–30, whiskers = 23 and 63; *S. sonnei* (male and female, respectively) minima = 0, maxima = 80 and 72, median = 4 and 7, IQR (bounds of box) = 2–12.8 and 3–28.5, whiskers = 29 and 69.* = Mann–Whitney *U*

(two-sided) *p* value = 0.007187, *S. flexneri* 2a male *n* = 128, female *n* = 129, and unknown sex = 3, *S. sonnei* male *n* = 129, female *n* = 122 and unknown sex = 2. WEC West Coast, CAT City of Cape Town, BOL Cape Winelands, OVE Overberg, CKA Central Karoo, GRO Garden State, CAC Sarah Baartman, CHA Chris Hani, NMM Nelson Mandela Bay, AMT Amathole, UKH Joe Gqabi, FRB Frances Baard, LEJ Lejweleputswa, MOT Manguang, SIS Harry Gwala, UTU uThukela, UMG uMhlanga, UGU =Ugu, AMJ Amajuba, ETH City of eThekwini, ILE iLembe, BOP Bojanala Platinum, KKA Dr. Kenneth Kaunda, WRA West Rand, SED Sedibeng, JBM City of Johannesburg, TSW City of Tshwane, EKH Ekurhuleni, GSI Gert Sibande, NKG Nkangala, EHL Ehlanzeni, WAT Waterberg. Maps in **A** were created using editable MapChart maps (https://www.mapchart.net/), edited and used according to the Creative Commons Attribution 4.0 International License (https://creativecommons.org/licenses/by-sa/4.0/). Map in **B** was adapted from image made available at Wikimedia Commons (https://commons.wikimedia.org/wiki/File: Map_of_South_Africa_with_district_borders_(2011).svg), and used in accordance to the Creative Commons Attribution 3.0 International License (https://creativecommons.org/licenses/by-sa/3.0/deed.en). Breakdown of isolate numbers are found in (**A** and **C**). Source data are provided as a Source Data file.

global context shows the epidemiology of *S. flexneri* 2a in South Africa is mainly driven by local transmission.

Bayesian analysis of population structure (BAPS) clustering, used to facilitate statistical analysis, grouped the *S. flexneri* subsample into four clusters (Sf clusters). Each Sf cluster was typically monophyletic among the study isolates and mostly in the global context, indicating sustained local transmission (referred to as endemic throughout), with the exception of Sf cluster 3 which grouped an apparently endemic sub-lineage (Sf cluster 3a, Fig. 2) with possibly imported strains (Sf cluster 3b, Fig. 2). The fifteen isolates in Sf cluster 3b were inferred as possibly imported by their genetic distance from the endemic lineages and close relatedness to global references (Fig. 2A).

Owing to their potential role in driving population dynamics, genes relating to AMR and virulence were characterised among the isolates. Partial-phenotyping and in silico genotyping confirmed MDR, resistance to three or more antimicrobial classes, is widespread in South African *S. flexneri* 2a (91.5% of isolates). All but one cluster (Sf cluster 1) were predominantly (≥98%) MDR (Fig. 2). Only 67% of Sf cluster 1 were MDR; all (*n* = 19) predicted pan-susceptible isolates belonged to the cluster and resistance to ampicillin, chloramphenicol,

tetracycline, and streptomycin was significantly lower relative to other clusters ($\chi^2$(1, *n* = 260) = 126.51, 191.70, 201.85 and 138.38, respectively; *p* = 0.0000 for all four antimicrobials) (Fig. 2B, Supplementary Data 1). Greater drug susceptibility due to the absence of the SRL, a MDR element encoding resistance to these four antimicrobials, was confirmed by poor mapping to the region (Fig. 2B, Supplementary Data 1).

Sf cluster 1 also had a distinct virulence profile, with a higher proportion of isolates containing *capU* and *traT* relative to other clusters, possibly indicating greater relative virulence (Fig. 2B, Supplementary Data 1). This is partially supported by the concentration of isolates derived from blood-culture (indicative of invasive infection) in this cluster (Fig. 2, Supplementary Data 1). The influence of pathogen factors on disease presentation is not well understood, but *traT* may have a role in innate immunity evasion[61], while *capU* was identified on a virulence plasmid associated with enteroaggregative *Escherichia coli*[62].

To further understand how the identified cluster characteristics have shaped shigellosis epidemiology in South Africa over time, we conducted temporal population dynamic modelling (implemented in BEAST, see methods). The previous emergence estimate for *S. flexneri* PG3 was 1848[13], while our *S. flexneri* 2a subsample likely emerged

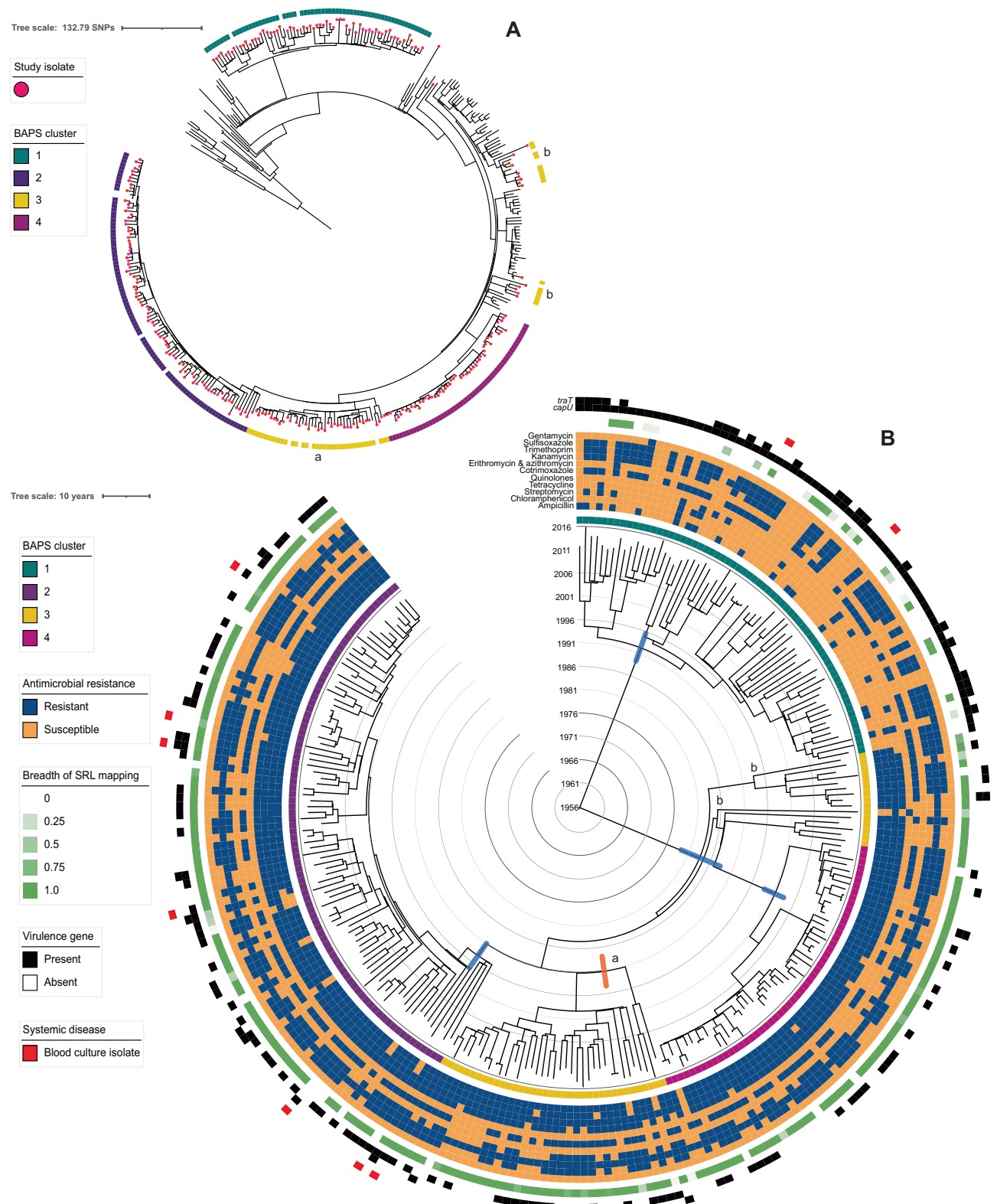

around 100 years later (1956; 95% HDP = late 1945 to late 1965). Our study emergence date instead coincides with the emergence (1948) of a rapidly expanding, intercontinental, PG3 subcluster[13]. The South Africa *S. flexneri* 2a population is split into two main Clades: 1) the reduced AMR Sf cluster 1 and 2) an MDR Clade comprising the remaining Sf clusters. Emergence dating suggests Sf cluster 1 established in South Africa in 1992 (95% HPD = mid-1989 to late 1995) after the MDR Clade (1983; 95% HPD = early 1979 to mid-1988) (Fig. 2B).

Thus, there was a successful introduction of a less AMR lineage into the country when MDR strains were already circulating, possibly aided by the increased presence of virulence genes in Sf cluster 1.

The other endemic clusters (Sf clusters 2, 3a and 4) all emerged during the 1990s; evidence of population diversification which is supported by the estimated increasing *S. flexneri* 2a population size between 2001 and 2009, (Supplementary fig. 3). Clusters 2 and 3a likely diverged from the same ancestral population in 1994 and 1991

**Fig. 2 | The contextualised population structure, antimicrobial resistance, and genetic features of South African *Shigella flexneri* 2a.** The maximum likelihood phylogeny **A** shows the *S. flexneri* 2a isolates South African sub-sampled isolates (indicated by pink dot overlying the terminal node) among Phylogroup 3 representatives. The BAPS clusters for the South African samples are shown in the outer ring according, coloured to the inlaid key. Sub-clusters of polyphyletic Cluster 3 are indicated with 'a' (locally circulating) and 'b' (possibly imported). **B** A Maximum Clade Credibility tree from Bayesian analysis shows the population structure of the *S. flexneri* 2a with branch lengths in time, scaled in years by the concentric rings radiating from the tree MRCA in the centre. BAPS clusters are indicated in the ring immediately adjacent to tree tips, coloured according to top inlaid key) and their MRCA 95% HPDs are indicated by blue bars for BAPS Clusters 1, 2, and 4. The circulating sub-clusters of polyphyletic BAPS Cluster 3 are labelled 'a' and 'b' and the MRCA 95% HPD of locally circulating 'a' indicated at the defining node in orange. Other metadata rings (from inner to outer, coloured according to the inlaid keys) show the: antimicrobial resistance profiles by antimicrobial class (combination of predicted and phenotypic resistance—see methods; blue = resistant and orange = susceptible), breadth of mapping coverage across the SRL (green), the presence of virulence genes *capU* and *traT* (black), and where an isolate was derived from a blood sample (rather than stool; red). MRCA most recent common ancestor, HPDs highest posterior density interval, BAPS Bayesian analysis of population structure, SRL Shigella resistance locus. Source data are provided as a Source Data file.

(95% HPDs: mid-1991 to early 1997, and late 1987 to mid-1994), respectively (Fig. 2). A long branch preceding the Sf cluster 4 MRCA suggests an evolutionary bottleneck, perhaps formed by migration of the strain out of South Africa then reintroduction in 2001 (95% HPD = early 1999 to late 2003) (Fig. 2B).

To determine the sub-national distribution of the identified genomic subtypes, we assessed the Sf cluster distribution across the country. Together, the Sf clusters were unevenly distributed between provinces (Fisher's exact $p = 0.0005$, Fig. 3). This suggests that co-existence of locally circulating strains in a country may feature sub-regional dominance by strains. We also found an association between Sf cluster and level of district urbanisation (Fisher's exact $p = 0.0001$) Owing to their potential role in driving population dynamics, genes relating to AMR and virulence were characterised among the isolates. Partial-phenotyping and in silico genotyping confirmed MDR, resistance to three or more antimicrobial classes, is widespread in South African S. *flexneri* 2a (91.5% of isolates). All but one cluster (Sf cluster 1) were predominantly (≥98%) MDR (Fig. 2). Only 67% of Sf cluster 1 were MDR; all ($n = 19$) predicted pan-susceptible isolates belonged to the cluster and resistance to ampicillin, chloramphenicol, tetracycline, and streptomycin was significantly lower relative to other clusters ($\chi^2(1, n = 260) = 126.51, 191.70, 201.85$ and $138.38$, respectively; $p = 0.0000$ for all four antimicrobials) (Fig. 2B, Supplementary Data 1). Greater drug susceptibility due to the absence of the SRL, a MDR element encoding resistance to these four antimicrobials, was confirmed by poor mapping to the region (Fig. 2B, Supplementary Data 1), perhaps reflecting genuine associations with level of urbanisation, or maybe a consequence of province associations.

The distribution of AMR reflected the distribution of the underlying Sf clusters. Free State province, predominantly Sf cluster 1, was negatively associated with MDR (9/13 isolates, −24%; Fisher's exact $p = 0.008$) (Owing to their potential role in driving population dynamics, genes relating to AMR and virulence were characterised among the isolates. Partial-phenotyping and in silico genotyping confirmed MDR, resistance to three or more antimicrobial classes, is widespread in South African S. *flexneri* 2a (91.5% of isolates). All but one cluster (Sf cluster 1) were predominantly (≥98%) MDR (Fig. 2). Only 67% of Sf cluster 1 were MDR; all ($n = 19$) predicted pan-susceptible isolates belonged to the cluster and resistance to ampicillin, chloramphenicol, tetracycline, and streptomycin was significantly lower relative to other clusters ($\chi^2(1, n = 260) = 126.51, 191.70, 201.85$ and $138.38$, respectively; $p = 0.0000$ for all four antimicrobials) (Fig. 2B, Supplementary Data 1). Greater drug susceptibility due to the absence of the SRL, a MDR element encoding resistance to these four antimicrobials, was confirmed by poor mapping to the region (Fig. 2B, Supplementary Data 1).

### *Shigella sonnei* population dynamics

We performed similar phylogenetic analyses on the South African *S. sonnei* subsample. The SNP-based maximum likelihood phylogeny contextualised the *S. sonnei* subsample ($n = 253$) within the global population (40 references), identifying the majority (98.4%) as Global Lineage III (Clade 3.7) isolates (Fig. 4A). We similarly found that the epidemiology of *S. sonnei*-attributable shigellosis was mainly driven by local transmission. We also found evidence population diversification, again starting in the 1990s (Fig. 4B, Supplementary Figs. 2 and 3). Most isolates (246/253) comprised three sub-clades of Lineage 3, Clade 3.7; sub-Clades 3.7.7, 3.7.9, 3.7.11 (Fig. 4A). Five strains were dispersed throughout the remainder of Clade 3.7 and seven were even more distantly related isolates (one Lineage 5, three sub-Clade 2.8.2, two Clade 3.6, and one Clade 3.4).

In common with *S. flexneri* 2a, partial phenotyping and in silico genotyping showed MDR was widespread in *S. sonnei* (93.7% of isolates). We also identified low-level AMR against two of the remaining widely effective antimicrobials, fluoroquinolones and third generation cephalosporins. Specifically, resistance against cephalosporins was phenotypically detected in two isolates, conferred by *blaCMY-2* and *blaCMY-4*. The only predicted fluoroquinolone-resistant (FQR) *S. sonnei* isolate, that contained triple mutations in the quinolone resistance determining regions (QRDR), (specifically *gyrA* S83L, *gyrA* D87G and *parC* S80I), was reported as phenotypically susceptible to ciprofloxacin. Single point mutations in *gyrA* ($n = 6$) were also detected in all five of the phenotypically quinolone-resistant isolates (Supplementary Data 1). One quinolone-resistant isolate and the QRDR triple mutant were from the Asia-associated sub-Clade 3.6.1 (Fig. 4)[63]. The remaining quinolone-resistant isolates were locally circulating strains, showing possible potential for de novo emergence of FQR strains in the country.

In contrast to *S. flexneri* 2a, we found no evidence of AMR profile or geographical distribution differences between circulating sub-Clades of *S. sonnei*, however, variation was seen in their virulence profiles. Most cluster-associated virulence genes were positively associated with Ss cluster 2 (Fig. 4B, Supplementary Data 1). Though this may be linked to the relatively higher presence of the pINV in Ss cluster 2, relative to other clusters (Pearson $\chi^2(1, n = 253) = 6.919, p = 0.009$) (Fig. 4B). Both *virF* and *ipaD* are known to be carried on the pINV[64,65], while *capU* was associated with pINV retention in the *S. sonnei* subsample ($n = 53; \chi^2 = 120.4, p = 0.0000$).

Bayesian population clustering, used again to aid statistical analysis, grouped the Lineage III isolates into three clusters (Ss clusters 1–3), the rest forming another cluster (Ss cluster 4), roughly corresponding with the *S. sonnei* subclade designations, with the exception of a 3.7 sub-Clade mis-identified as 2.8.2 (Fig. 4). Bayesian phylogenetics highlighted that the most recent common ancestor of all circulating *S. sonnei* in South Africa emerged in 1967 (95% HPD = late 1957 to early 1978) (red arrow, Fig. 4B). The circulating strain likely diversified into the identified sub-Clades in the 1990s. All emerging around the same time, sub-Clades 3.7.11 and 3.7.9 (Ss clusters 1 and 3, respectively) and the two sub-Clades of Ss cluster 2 (3.7.7 and the mis-identified 2.8.2 sub-Clade within Clade 3.7) likely emerged in 1996 (95% HPD = mid-1993 to late 1999), 2001 (95% HPD = late 1998 to late 2003), 1998 (95% HPD: 1996–2001), and 1996 (95% HPD: 1993-1998), respectively (Fig. 4). Population diversification in *S. sonnei* was also supported by an estimated increase in population size, though the increase was likely later than in *S. flexneri* 2a from around 2007–2010 to 2013

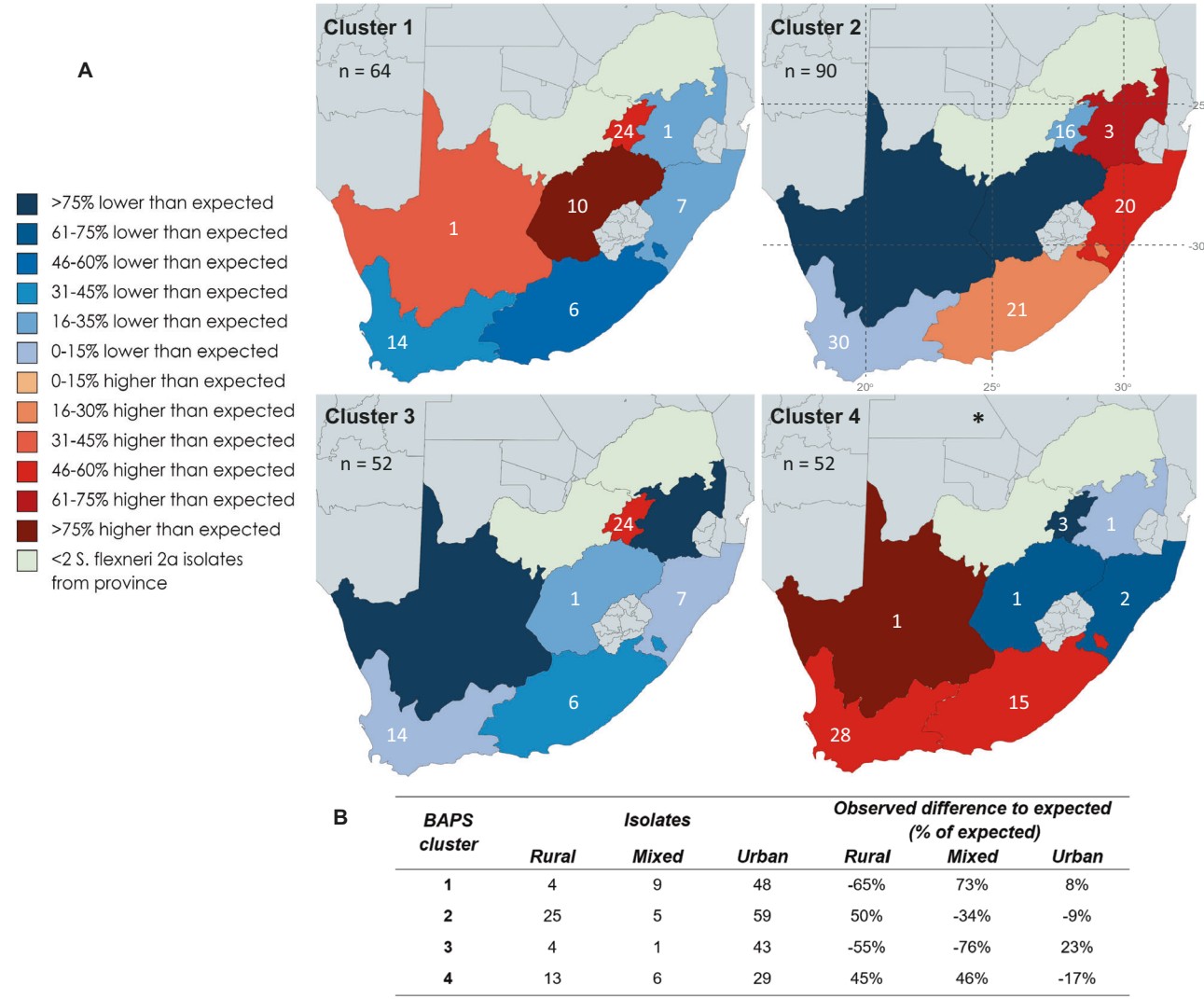

**Fig. 3 | Geographic distribution of *S. flexneri* 2a BAPS clusters within South Africa.** Each cluster has a distinct distribution by province across South Africa. **A** The under or over representation of each BAPS cluster by province based on the total number of isolates from each province; blue = fewer isolates than expected, red = more isolates than expected) (percentage difference determined using: $\frac{O-E}{E}$, where *O* = actual number of cluster isolates and *E* = expected number of cluster isolates based on the total number of sample set isolates). Individually, Sf cluster 4 is statistically associated with Northern Cape, Western Cape, and Eastern Cape. Latitude and longitude are overlaid on BAPS cluster 2 map. **B** The presence of each cluster also varies by level of urbanisation, defined at the district level. * Fisher's exact *p* value *p* = 0.0005. Source data are provided as a Source Data file. Maps in **A** were created using editable MapChart maps (https://www.mapchart.net/), edited and used according to the Creative Commons Attribution 4.0 International License (https://creativecommons.org/licenses/by-sa/4.0/). BAPS Bayesian analysis of population structure.

| BAPS cluster | Isolates | | | Observed difference to expected (% of expected) | | |
|---|---|---|---|---|---|---|
| | Rural | Mixed | Urban | Rural | Mixed | Urban |
| 1 | 4 | 9 | 48 | -65% | 73% | 8% |
| 2 | 25 | 5 | 59 | 50% | -34% | -9% |
| 3 | 4 | 1 | 43 | -55% | -76% | 23% |
| 4 | 13 | 6 | 29 | 45% | 46% | -17% |

(Supplementary Fig. 3). Our *S. sonnei* Bayesian modelling dating estimates are consistent with prior estimates, with our estimated MRCA date for all *S. sonnei* subtypes (1786; 95% HPD = mid-1714 to early 1849) (Fig. 4B), overlapping with the prior MRCA estimate for all *S. sonnei* (median = 1669, 95% HPD = 1554–1763)[12].

## Discussion

This study has highlighted several distinctions between the epidemiology of *S. flexneri* 2a and *S. sonnei* in South Africa. The four circulating clusters of *S. flexneri* 2a varied in their AMR and virulence profiles while the phylogenetically identifiable, circulating subClades of *S. sonnei* had little variation. The lack of evidence for MSM-associated Lineages is believed to be due to the high prevalence of shigellosis in the general population in South Africa, relative to nations where MSM-associated Lineages are observed. This high endemic transmission is likely to have obscured the signal of MSM-associated transmission rather than a lack of MSM-associated transmission.

The diversity of *S. flexneri* 2a clusters, and presence of all clusters across the whole study period (Supplementary fig. 4), supports strain co-existence, previously observed on a global level[13]. Contrastingly, clonal replacement in *S. flexneri* was found on a national level before in countries in South East Asia[17] though this occurred over a 5–20 year period, so our study may not have covered a sufficient time period to observe clonal replacement[17]. We found no evidence that the introduced, reduced AMR *S. flexneri* 2a Lineage was replacing the already present MDR Clade. However, on a regional (Southeast Asia) and global level, clonal replacement in *S. flexneri* is less obvious; with re-emergence of strains after their apparent replacement[13,17]. The epidemiological dynamics of *Shigella* in South Africa may be more like the regional dynamics identified in the Southeast Asian study[17] as South Africa is a comparable land mass to the total Southeast Asian region studied. This is supported by the distinct provincial distributions of the identified circulating *S. flexneri* 2a clusters observed in this study.

The AMR profile similarity and geographic homogeneity of *S. sonnei* clusters are consistent with clonal replacement observed in

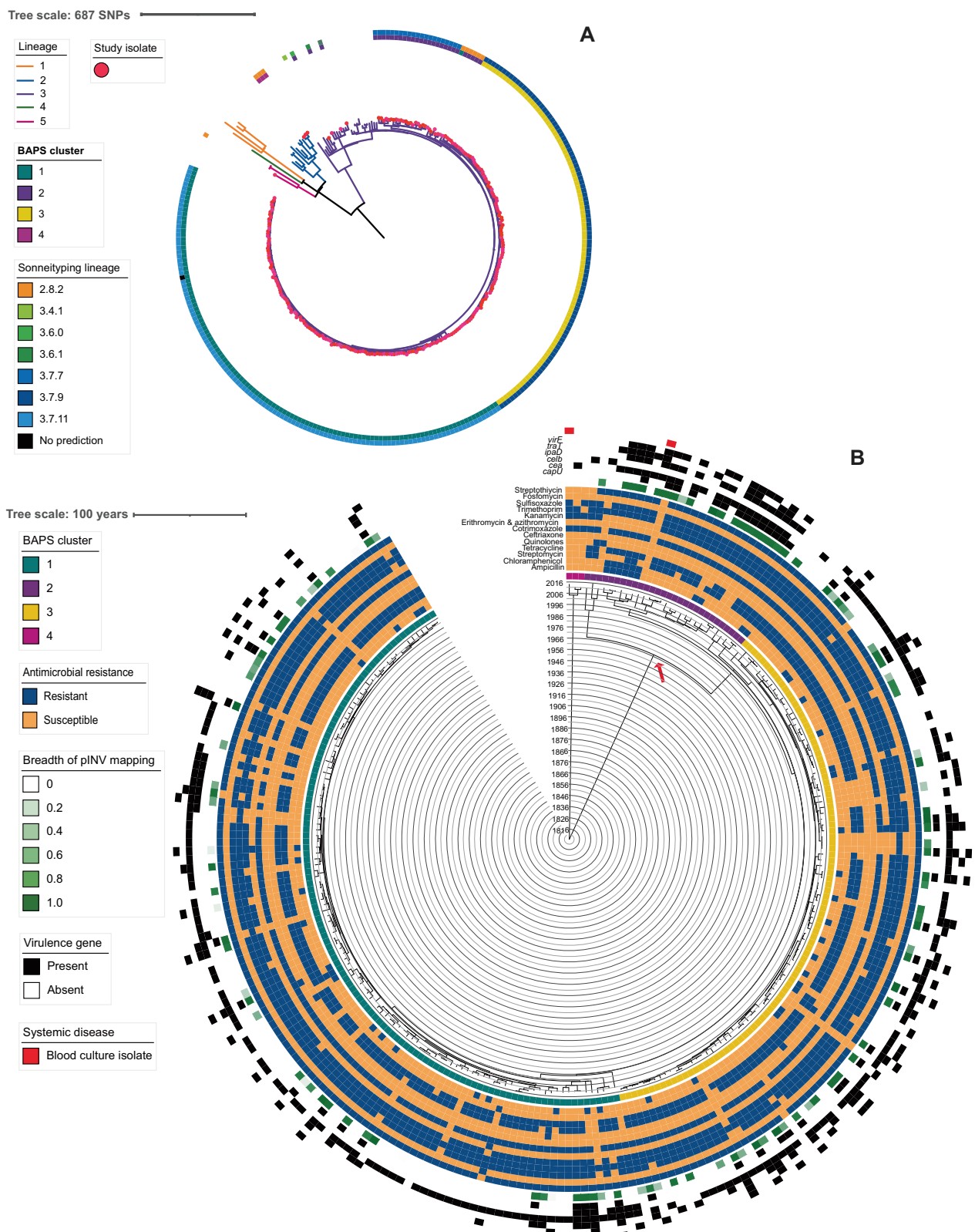

previous global and national (Vietnam, China) studies of *S. sonnei*[12,66–68]. Greater pINV retention was seen in the smaller circulating *S. sonnei* sub-Clade (3.7.7) despite having emerged around the same time as the other sub-Clades (3.7.9 and 3.7.11) (1996–2001), possibly indicating a link between epidemiological success and pINV stability. Reduced stability of the pINV in *S. sonnei*, compared to other *Shigella*, is likely a lifestyle adaptation towards becoming reliant on human to human transmission[21]. The presence of all clusters across the study period (Supplementary fig. 4), however, does not support clonal replacement driven by differences in pINV retention. While it is clear further study is needed to confirm a link between pINV stability and epidemiological success, our study supports pathogen ecology, in particular lifestyle, drives *Shigella* epidemiology.

**Fig. 4 | The contextualised population structure, antimicrobial resistance, and genetic features of South African *Shigella sonnei*.** The maximum likelihood phylogeny **A** shows the *S. sonnei* sub-sampled isolates (indicated by pink dot overlying the terminal node) among some global representatives. The previously defined Lineages are indicated by branch colour while the outer ring shows the predicted genotype for the study isolates, predicted by *Sonnei*Typing software, coloured according to inlaid key. **B** A Maximum Clade Credibility tree from Bayesian analysis shows the population structure of the S. *sonnei* with branch lengths in time, scaled in years by the concentric rings radiating from the tree MRCA in the centre. BAPS clusters are indicated in the ring immediately adjacent to tree tips, coloured according to the top inlaid key. Other metadata rings (from inner to outer, coloured according to the inlaid keys) show: antimicrobial resistance profiles by antimicrobial class (combination of predicted and phenotypic—see methods; blue = resistant and orange = susceptible), breadth of mapping coverage across the pINV (green), presence of virulence genes *capU, cea, celb, ipaD, traT, virF* (black), and derived from a blood sample (rather than stool; red). MRCA for endemic clusters indicated by red arrow. BAPS Bayesian analysis of population structure, *pINV* large virulence plasmid. *MRCA* most recent common ancestor, *HPDs* highest posterior density interval, *BAPS* Bayesian analysis of population structure. Source data are provided as a Source Data file.

Our observed epidemiological differences between *S. sonnei* and *S. flexneri* 2a add to the existing evidence of serogroup-specific lifestyles. Prior evidence includes variation in AMR-linked epidemiological success; clonal replacement and intercontinental dissemination are driven by acquisition of AMR in *S. sonnei* but not *S. flexneri*, except amongst men who have sex with men (MSM)[13,67,69,70]. Epidemiological success may in MSM be linked to AMR regardless of serotype, promoted by, almost exclusively, direct contact transmission and high rates of antimicrobial use among the sub-population of affected MSM[71–73]. Here, we have found a reduced AMR lineage (Sf cluster 1) was successfully introduced to South Africa in the early 1990s when MDR strains were already present highlighting that AMR is not always necessary for *S. flexneri* strain success. Loss of the SRL has been reported previously[27], as has the establishment of a drug-susceptible lineage of *Salmonella enterica* subspecies enterica, serovar Typhimurium, in the presence of MDR strains circulating in Malawi[74].

An important finding of this study was the evidence of population diversification, in both serotypes, beginning in the 1990s. The emergence and introduction of new *Shigella* strains in the country occurred during the 1990s coinciding with the end of Apartheid (1992–1994) and the beginning of the HIV epidemic in the country[75], both of which may have driven changes in shigellosis epidemiology. Regarding post-Apartheid influences, changes in cross-border travel during this time may be linked to the introduction of Sf cluster 1 into South Africa in 1992 and promoted *S. flexneri* 2a population expansion through an influx of immunologically naïve hosts. However, population expansion and strain diversification could also be linked with increasing HIV prevalence. Strain divergence in *S. sonnei* (clusters 1 and 3) and *S. flexneri* 2a (clusters 2 and 3a) occurred from 1991 to 2001, coinciding with the peak of new HIV infections (2000 and 2001); the number of people living with HIV in South Africa continues to increase[75] (Supplementary fig. 3). The influence of HIV is supported by previous research, as HIV is a risk factor for both shigellosis infection and systemic disease[23,24,76–81]. Furthermore, similar dynamics have been observed in *Salmonella* and tuberculosis (TB). Specifically, invasive *Salmonella* Typhimurium in SSA, was predominantly caused by two Lineages[82] whose clonal expansions were linked to peaks in HIV prevalence and the expansion of HIV-1[82], and HIV is a known driver of the TB epidemic[83,84].

While this study is unable to determine the relative importance of the end of Apartheid and the HIV epidemic on shigellosis in South Africa, understanding the link between HIV and shigellosis is particularly important for effective public health policy. Particularly as the disparity in the average age of infection between males and females, identified in this study, may additionally be partially due to the greater numbers of women over the age of fifteen years living with HIV in South Africa[75] (rather than child care responsibilities as explored above). Another limitation of this study relates to the paucity of surveillance data for gastrointestinal illnesses in endemic regions. Furthermore, the sample comes only from people attending public hospitals and there is low sampling coverage in rural areas. However, we have tried to mitigate this by carefully selecting our subsample relative to case reporting data, but the potential for sampling bias to have impacted our overall findings cannot be completely ruled out.

Our study is the first detailed examination of shigellosis genomic epidemiology in an African nation and shows that shigellosis epidemiology in South Africa is serotype specific. Our findings offer insights into likely drivers of *Shigella* evolution and epidemiology, supporting pathogen lifestyle as a driver of shigellosis epidemiology, explaining the observed serogroup differences in AMR trends and pINV stability in this study, and globally. We also highlight the potential importance of the HIV epidemic on shigellosis and underline the importance of further research into the relationship between HIV and shigellosis. Based on the findings of this study, we suggest that serogroup-specific public health approaches, which take HIV into account, may be needed for effective shigellosis control nationally, across SSA, and globally.

## Methods

### Ethics statement
No individual consent was required or sought as all isolates were collected as part of routine surveillance and ethical approval for the use of patient data for public health activities was granted by the Human Research Ethics Committee of the University of the Witwatersrand (Protocol Numbers: M060449 and M110499).

### Sample collection, sub-sampling, and sequencing
We created a sub-sample ($n = 561$) from all the biochemically identified *Shigella flexneri* 2a ($n = 286$) or *Shigella sonnei* ($n = 275$) collected during routine surveillance (of all age groups) in South Africa (2011–2015) carried out by The Group for Enteric, Respiratory and Meningeal Diseases Surveillance in South Africa (GERMS-SA)[54–58]. Total *Shigella* isolated during the study period was 8251, of which 2287 were *S. flexneri* 2a and 1841 were *S. sonnei*. Isolates were randomly sub-sampled across those collected throughout the study period. Random sub-sampling was achieved by selecting every 8th (*S. flexneri* 2a) or 5th (*S. sonnei*) isolate, based on SA lab number, from the database of collected, surveillance isolates (collected from any body site, 1 January 2011–31 December 2015, for which demographic data was available), up to a minimum of 250 isolates. SA lab number was assigned upon arrival at the reference laboratory, from a surveillance site. If selected isolate could not be located, then the previous isolate in the database was selected instead. All surveillance sites were public hospitals.

All study isolates were whole genome sequenced using Illumina HiSeq 4000 sequencing equipment and the DNA library was prepared using the Illumina Nextera XT DNA Library Prep Kit (Illumina, FC-131-1096)[85]. The quality of some of the isolate sequences (*S. flexneri* 2a $n = 119$ and *S. sonnei* $n = 65$) was poor (due to GC library bias caused by the transposon used for tagmentation) and were subsequently re-sequenced at the Centre for Genomic Research (CGR, University of Liverpool) using the Illumina NovaSeq 6000 platform; the DNA library was constructed using the NEBNext Ultra II FS DNA Library Prep Kit for Illumina[16]. Re-sequenced reads were pooled with the original isolate reads prior to quality trimming.

Two separate public data sets were used as references for the maximum likelihood phylogenetics; one for *S. flexneri* 2a ($n = 286$) and one for *S. sonnei* ($n = 40$) (accession numbers available in Supplementary Data 1). For, both sample sets, isolates were selected from

across the known global Phylogeny, with care taken to include isolates from all genomic subtypes (e.g., Phylogroups/Lineages). A subset of the *S. flexneri* 2a reference sample set (*n* = 116) was used to generate a Phylogroup 3 phylogeny, and exclusively contained all the Phylogroup 3 isolates from the larger reference sample set.

## Data collection

National surveillance involved a network of public hospitals and laboratories from across all nine provinces, further details can be found in GERMS-SA annual reports[54–58]. *Shigella* isolates were identified as part of active laboratory-based surveillance for bacterial causes of diarrhoea. Patients presenting at health-care facilities with a history of diarrhoea defined as three or more loose stools per day, either with or without blood, were eligible to have a stool specimen submitted for microscopy and culture for bacterial pathogens; final decision on stool specimen submission rested with the treating healthcare practitioner. All positive bacterial cultures were forwarded to CED for further characterisation, according to standard characterisation procedures. Some patient data was collected during surveillance, including age and sex (based on medical records); no records were collected on sexuality. A single isolate only was collected from each patient sample. Isolates came from 257 males (128 *S. flexneri* 2a and 129 *S. sonnei*) and 151 females (129 *S. flexneri* 2a and 122 *S. sonnei*).

Recorded case numbers, taken from the GERMS-SA annual reports, were used to examine the background epidemiology shigellosis in the country, and the representation of the larger surveillance dataset by our study isolates[54–58]. Due to missing data from the GERMS-SA annual reports, comparisons were made using data collected during 2011 to 2013.

Each province in South Africa is sub-divided into district municipalities and metropolitan municipalities, both referred to as districts throughout and categorised by degree of urbanisation according to the methods laid out by the European Commission in March 2020 (https://ec.europa.eu/16urostat/cros/system/files/bg-item3j-recommendation-e.pdf). Densely populated districts are referred to, here, as 'urban', intermediate density as 'mixed', and thinly populated as 'rural'.

## Quality control

All raw sequence reads were quality trimmed with Trimmomatic (v0.38) and SeqTK (v1.3) (https://github.com/lh3/seqtk)[86]. In addition to quality trimming, a further eight bases were removed from the ends of all South African study isolate reads with SeqTK due to poor quality.

Quality was assessed before and after trimming with FastQC (v0.11.8) and MultiQC (v1.5)[87,88]. Read mapping quality was assessed using Qualimap (v2.2.2-dev)[89]. Genome assembly quality was assessed using Quast (v8.13) against the read mapping reference genome[90,91]. All included isolates had sequences with: (1) per base mean Phred score ≥28, (2) per sequence Phred score ≥28, (3) even per base sequence content, (4) even per sequence GC content curve and peak GC content between 48–54%, 5) per base N content <5, (6) < 5% overrepresented sequences, and (7) < 5% sequences with adaptor sequences.

No sequencing data was received for thirteen *S. flexneri* 2a and five *S. sonnei*. Twenty *S. flexneri* 2a isolates were excluded from further analysis: seven for poor sequence quality, two for uneven per base sequence content, five for uneven GC content, two for poor mapping against the complete *S. flexneri* 2a reference genome (<20 mean read depth), and four for being mis-identified as *S. flexneri* 2a.

Seventeen *S. sonnei* were excluded from further analysis: four for uneven per base sequence content, seven poor mapping to the *S. sonnei* reference genome (<20 mean read depth), and six for being misidentified as *S. sonnei*. All unpaired reads for both serotypes were also excluded from further analysis due to poor per base sequence content. Of the included 260 *S. flexneri* 2a and 253 *S. sonnei* isolates, 111

and 57 were re-sequenced, respectively. All unpaired reads were excluded due to poor per base sequence content.

## In silico strain typing

In silico species confirmation and prediction of serotype was done using ShigaTyper (v1.0.6) (https://github.com/CFSAN-Biostatistics/shigatyper) and *Sonnei*Typing *sonnei*_genotype.py (v1) (https://github.com/katholt/sonneityping) was used to predict genotype of *S. sonnei* isolates[63]. Both software tools were run using isolate sequence reads; raw reads for ShigaTyper but trimmed reads for *Sonnei*Typing.

## Maximum likelihood phylogenetics

All maximum likelihood phylogenetic trees were generated from core-SNP alignments using RaxML-NG (v0.6.6; GTR + G substitution model, 1000 bootstrap validation and mid-point rooted)[92]. Core-SNP alignments (*S. flexneri* 2a: all Phylogroups phylogeny = 34,768 SNPs and Phylogroup 3 phylogeny = 13,279 SNPs, *S. sonnei:* 6870 SNPS) were generated from quality-trimmed study isolate sequence reads and phylogenetic reference isolate sequence reads. Sequence read mapping, performed with bwa mem (v0.7.17), was to either the *S. flexneri* 2a 301 strain complete genome (accessions: NC_004337.2, NC_004851.1) or the *S. sonnei* 53 G strain complete genome (accessions: HE616528.1, HE616529.1, HE616530.1, HE616531.1, HE616532.1). Samtools (v1.9), samclip (v2.27.1) and Picard (v2.23.8) (https://broadinstitute.github.io/picard/) were used to sort, filter and index mapped reads. Variant calling was performed with samtools mpileup (v1.9) and bcftools (v1.9)[93–95]. Consensus sequences were defined with samtools vcfutils.pl (min. depth = 4), with plasmids, mobile genetic elements, and phaster (https://phaster.ca/) identified phage sequences masked with bedtools (v2.27.1)[93–98]. Masked consensus sequences were run through gubbins (v2.4.1)[99].

## Bayesian phylogenetics

Serotype-specific time trees were generated with Bayesian Evolutionary Analysis by Sampling Trees software (BEAST2) (v2.6.3). Generated from study isolate only core-SNP alignments (*S. flexneri* 2a: 7936 SNPs, *S. sonnei:* 4161 SNPs), created using same methods as for maximum likelihood phylogenetics. The fasta alignment was converted to nex with seqmagick (v0.8.0) (https://github.com/fhcrc/seqmagick/).

Two isolates were excluded from each serotype-specific BEAST phylogeny due to being outliers for the molecular clock signal, visually identified in TempEST (v.1.5.3) using RaxML-ng (bootstraps, GTR + G model) generated, study isolate-only, serotype-specific, maximum likelihood phylogenies[92,100]. Once outlier isolates were excluded, estimated (in TempEST) molecular clock rates were 3.1921e-4 and 6.0361$e^{-4}$ substitutions per site per year (correlation co-efficient = 0.27 and 0.39) for *S. flexneri* 2a and *S. sonnei*, respectively.

We used an extended coalescent Bayesian skyline tree model, with a relaxed clock and a log normal prior distribution (prior rate = 1e-6), and site model averaging with BmodelTest, transition-transversion split (prior mutation rate = 1.0), all other priors were left as default[101].

For *S. flexneri* 2a we ran a single MCMC chain with a length of 5,043,000,000 (sampling every 1,000,000), 39% burn-in was removed. While for *S. sonnei* we ran three MCMC chains with a length of 3,300,000,000 (sampling every 100000), removing around 67% burn-in from all three while combining with logcombiner. The tree topology was generated with treeannotator (v2.6.3) and visualised in FigTree (v1.4.4) (https://github.com/rambaut/figtree) and the interactive tree of life (ITOL) online platform[102]. A minimum effective sample size (ESS) of 200 was achieved for all model parameters, checked using Tracer (v2.6.3)[103].

Study isolates population clusters were defined using RhierBAPS (v1.1.3) and serotype-specific, study isolate only core-SNP alignment in Rstudio (v1.4.1717; R v4.1.0)[104].

## Genome assembly and annotation

Draft genomes were assembled using unicycler (v0.4.7) and quality-trimmed sequence reads[105]. Genome annotation was achieved with Prokka (v1.14.5) against the *Escherichia* database.

## Antimicrobial resistance profiling

AMR profiles were defined based on phenotypic data, where available, or in silico prediction. Gene determinants were identified by starAMR (v0.5.1), or AMRfinderPlus (v3.2.3) for chloramphenicol resistance in *S. flexneri* 2a (due to better accuracy against phenotypic data). (Fluoro)quinolone resistance conferring point mutations in the QRDR were detected in silico with *Sonnei*Typing (*S. sonnei*) or manual extraction of amino acids in *gyrA* at positions 83 and 87 and *parC* position 80 (*S flexneri* 2a). Partial phenotypic data was available for ampicillin, chloramphenicol, streptomycin, tetracycline, nalidixic acid, ciprofloxacin, trimethoprim, and ceftriaxone, and are detailed in the Supplementary Data 1 alongside the detection of genotypic markers.

Accuracy of the in silico AMR phenotype predictions were assessed against recorded phenotype. Accuracy by antimicrobial, for both AMRfinder and starAMR, was defined as number of correct resistant phenotype predictions and number of correct susceptible phenotype predictions. Association with resistance was also assessed for each antimicrobial resistance determinant, using genotype. A minimum association of 50% was required for a resistance determinant to be accepted as associated with resistance and, if present, used to predict a resistance phenotype. In *S. sonnei*, *aadA1* or *aadA2* alone were poor predictors of streptomycin resistance, though if present together the association was strong enough that a resistant phenotype was predicted. In *S. flexneri* 2a, *aadA2* was not used to predict resistance to streptomycin.

## Virulence profiling

A virulence profile was generated for all study isolates by comparing the draft genome assemblies against the VirulenceFinder database, using VirulenceFinder (v2.0.4–1), and against a local, curated virulence genome database, based off a previous study (Supplementary Data 1), with BLASTn (v2.10.0+)[33]. Only genes with ≥99% sequence identity and database gene coverage were accepted.

The presence of the SRL-PAI, SRL and pINV was assessed by mapping isolate reads to these known virulence loci, within a reference genome. Quality-trimmed reads were mapped to various reference genomes, known to contain the relevant loci, with bwa mem (v0.7.17) (Supplementary Data 1). Read mapping of in silico generated positive and negative controls were also assessed, using random reads (paired, 100-70 bases, insert size 150-300) generated from positive and negative control genomes with the bbmap randomreads.sh script (v38.00) to 60x coverage (www.sourceforge.net/projects/bbmap/) (Supplementary Data 1). Reads converted into a set of forward and a set of reverse reads with seqkit (v0.10.1)[171].

## Statistics

Most associations were compared with a chi-squared test of association, carried out using raw numbers and reported, in most cases, as a percentage difference. For background epidemiology (using data extracted from GERMS-SA annual reports[54–58]) and sample set representativeness, percentage differences are defined as the observed cases or samples (O) as a fraction of the national total (T) minus the expected (E) fraction of the national total ($\frac{O}{T_o} - \frac{E}{T_c}$), where $T_o$ is the total observations (cases or samples) and $T_c$ is the total reported cases. For associations between isolate or patient metadata, percentage

difference is defined as the difference between observed and expected isolate numbers as a fraction of the expected number of isolates ($\frac{O-E}{E}$).

Associations between AMR determinant, or profile, and geographic region was tested with a Fisher's exact test. Chi-squared tests were used to examine the associations between AMR determinants, or profile, and *Shigella* sub-population. Meanwhile, associations between

**Table 1 | Number of sequenced samples from each province and district by serotype**

|  | *Shigella flexneri* 2a | *Shigella sonnei* | Total |
|---|---|---|---|
| Eastern Cape | 48 | 32 | 80 |
| Amathole DM | 24 | 14 | 38 |
| Chris Hani DM | 1 |  | 1 |
| Nelson Mandela Bay MM | 18 | 16 | 34 |
| Joe Gqabi DM | 2 |  | 2 |
| Sarah Baartman DM |  | 1 | 1 |
| District unknown | 3 | 1 | 4 |
| Free State | 13 | 13 | 26 |
| Lejweleputswa DM |  | 4 | 7 |
| Mangaung MM | 10 | 9 | 19 |
| Gauteng | 64 | 134 | 198 |
| City of Johannesburg MM | 23 | 64 | 87 |
| City of Tshwane MM | 16 | 27 | 43 |
| Ekurhuleni MM | 10 | 21 | 31 |
| Sedibeng DM | 6 | 7 | 13 |
| West Rand DM | 4 | 4 | 8 |
| City of Tshwane MM |  | 1 | 1 |
| District unknown | 5 | 10 | 15 |
| KwaZulu-Natal | 36 | 29 | 65 |
| Amajuba DM |  | 4 | 4 |
| iLembe DM | 1 |  | 1 |
| Ugu DM | 2 | 3 | 5 |
| uMgungundlovu DM | 5 | 4 | 9 |
| City of eThekwini MM | 24 | 16 | 40 |
| Harry Gwala DM | 1 | 1 | 2 |
| Uthukela DM | 2 |  | 2 |
| District unknown | 1 | 1 | 2 |
| Limpopo | 1 |  | 1 |
| Waterberg DM | 1 |  | 1 |
| Mpumalanga | 5 | 8 | 13 |
| Ehlanzeni DM | 2 | 6 | 8 |
| Gert Sibande DM | 3 | 1 | 4 |
| Nkangala DM |  | 1 | 1 |
| Northern Cape | 3 | 1 | 4 |
| Frances Baard DM | 3 | 1 | 4 |
| North West | 1 | 4 | 5 |
| Bojanala Platinum DM | 1 | 1 | 2 |
| Dr Kenneth Kaunda MM |  | 3 | 3 |
| Western Cape | 89 | 32 | 121 |
| Cape Winelands DM | 2 | 1 | 3 |
| Central Karoo DM | 1 |  | 1 |
| City of Cape Town MM | 69 | 26 | 95 |
| Overberg DM | 1 |  | 1 |
| West Coast DM | 8 | 2 | 10 |
| Garden Route DM | 4 | 2 | 6 |
| Unknown district | 4 | 1 | 5 |

*MM* metropolitan municipality, *DM* district municipality.

virulence gene and *Shigella* sub-population, were assessed using odds ratios and two-tailed *Z* tests (Supplementary Data 1). The Bonferroni correction was used to adjust the statistical significance threshold for multiple comparisonsTable 1.

## Reporting summary
Further information on research design is available in the Nature Portfolio Reporting Summary linked to this article.

## Data availability
The study isolates' raw sequences have been deposited in the European Nucleotide Archive under project accession PRJEB55173 and individual isolate accession numbers can be found in the Supplementary Data 1. Accession numbers for reference isolates used in the study are also provided in the supplementary table. All patient metadata, and all data generated from the analysis of this metadata, used in this study are provided in the Supplementary Data 1 and Source Data files, respectively. Source data are provided with this paper.

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

## Acknowledgements

We thank all participants of the NICD GERMS-SA Laboratory Surveillance Network for submission of clinical isolates of *Shigella* species to the NICD. The surveillance network includes laboratories belonging to the Department of Health (NHLS laboratories) and laboratories that form part of the private sector. Next-generation sequencing and library construction were delivered via the BBSRC National Capability in Genomics and Single Cell Analysis (BB/CCG1720/1, N.H.) at Earlham Institute, by members of the Genomics Pipelines Group. The project was supported by both a Global Challenges Research Fund (GCRF) data & resources grant BBS/OS/GC/000009D (N.H.) and the BBSRC Core Capability Grant to the Earlham Institute BB/CCG1720/1 (N.H.) and Core Strategic Programme Grant BBS/E/T/000PR9817 (N.H.). This work was also supported by Medical Research Council grant MR/R020787/1 (K.B.). For the purpose of open access, the author has applied a Creative Commons Attribution (CC BY) licence to any Author Accepted Manuscript version arising from this submission.

## Author contributions

Conceptualisation: K.S.B., K.K. Data curation: K.S.B., N.H., G.E.S. Formal analysis: G.E.S. Funding acquisition: K.S.B., M.I.G. Investigation: G.E.S. Methodology: G.E.S., R.J.B. Project administration: G.E.S. Resources: N.H., K.K., A.S., J.T. Supervision: K.S.B., M.I.G., A.S., J.T., K.K. Visualisation: G.E.S. Writing—original draft: G.E.S. Writing—reviewing and editing: all authors.

## Competing interests

The authors declare the following competing interests: M.I.G. is currently an employee of GSK, but this work was completed prior to her joining. The remaining authors declare no competing interests.
