## [Peer Review File · Nature Communications]

The genomic epidemiology of shigellosis in South AfricaREVIEWER COMMENTS

Reviewer #1 (Remarks to the Author):

The authors present a very valuable study into the genomic epidemiology of Shigella in south Africa. They report new insights, especially regarding the difference in epidemiology of the different species. Technincally the study is sound. Few comments that might improve the paper:

1. The authors talk about "serotype" specific epidemiology; howeve, to my knowlegde the authors compare two different Shigella species (flexneri and sonnei) rather than two serotypes. Throughout the manuscript this is creating confusing.
2. In several parts of the manuscript the authors talk about "ecology" (ecology as a driver of shigellosis epidemiology). It is unclear what the authors exactly mean by this and needs further explanantion.
3. It is not clear to what extent patient information is available. It would be high beneficial for to know the difference in incidence of both species in MSM versus non-MSM. it might also help to explain the difference in population ecology, and why amr is not necessary for flexneri success (probably because it finds its niche in non-msm more with lower antimicrobial use?)
4. Extended data figure 3: why is it possible for sonnei to separate between different age-groups and gender, and not for flexneri?

Reviewer #2 (Remarks to the Author):

GENERAL: This is a nicely presented article on the genomic epidemiology of Shigella infections in South Africa. The manuscript was very clear and easy to read, although there were some typographical errors throughout. The genomic comparison between Sh. sonnei and Sh. flexneri in the South African context was very interesting. I thought that the BEAST analysis was good showing timescales for potential introduction of strains into South Africa. The data are novel in that they come from surveillance data over a longer period of time than other studies conducted in Africa, such as the GEMS study.

Overall, the methodology was sound, based on genomic analysis of isolates collected through a surveillance program, although I did think that the statistical analysis of the geographic and temporal trends did not take into account the way the the data were collected. What were the potential biases introduced into the data through the collection methods of the surveillance? It is disappointing that there wasn't good epidemiological data associated with the study isolate. This led to assumptions and speculation in some places in the manuscript, which were never acknowledged as a limitation.

SPECIFIC:

L107: Doesn't this just indicate that there may have been outbreaks of these strains in these provinces in these time periods? It doesn't really go to the problem of sampling bias from the surveillance system

itself, which is the real problem that is difficult to account for.

L129: Just to note that you haven't presented average ages, but medians and IQRs.

L136: I don't think that the reference to the Orthodox Jewish community is necessary given that we have always known that caring for a patient with shigellosis is a risk factor for infection.

Line 182: Note missing reference

Line 245: Note missing reference

Line 290: Wording is clumsy ('...supported by estimated population increase population size...'). I am not sure what is meant here.

Line 307: Strains is spelt incorrectly 'stains'.

Line 309: The comparison of South Africa with South East Asia was confusing as South East Asia isn't the same geographical size as South Africa. Were you meaning to compare your study findings with that of Chung et al (ref #17)? I might have missed something here.

L325: I think it is speculation to say that 'epidemiological success in MSM is perhaps linked to AMR...'. I don't think we can really know the reason for persistence of these MDR strains in MSM from genomics studies alone.

L331: I don't think you can claim that there hasn't been introduction of less drug-susceptible Shigella strain, as countries see persistence of strains with different characteristics all the time, they just don't publish it.

L364: What is the total number of biochemically identified Shigella and Sh. flexneri and sonnei? It is important for readers to understand what proportion your sample makes up of the population of isolates.

L366: It would help the reader to briefly explain the GERMS-SA surveillance and how isolates were collected and from whom. This could be done in more detail in L389. I just don't get a feel for who gets tested for Shigella and when. What kind of undercount was there in surveillance data?

L390: It is important to discuss in the manuscript that the observed differences relate to the nature and performance of surveillance.

L510: Where does the national total come from? I really didn't get a feel for national rates of shigellosis in South Africa either.

L514: You mention using odds ratios although I couldn't see them anywhere in the paper. If you have used them, it would be better to present them with 95% confidence intervals than p values. I am not convinced of the need of Bonferroni correction.

Reviewer #3 (Remarks to the Author):

This is an excellent manuscript describing the genomic epidemiology of Shigella in South Africa, marking an important contribution to the dearth of evidence around Shigella epidemiology in the sub-Saharan Africa continent. Most impressive in my mind, is this enteric surveillance system that is set up in South Africa providing an incredibly rich isolate and data repository enabling this type of analysis.

That age of infection was older in females than in males is interesting, and to the best of my knowledge, this has not been demonstrated previously. In addition to caregiving responsibilities, could this also be

explained by the surveillance system and/or culturability of the isolates? For example, it's been shown that Shigella isolate recovery is more common in older children (ie culture negative/ PCR positive infection more common in young children). Shigella can be tricky to recover, and is impacted by transport media etc. Could it also be that surveillance facilities with more female participants had a longer transport time to the lab?

The authors conclude that the new Shigella strains in the 1990s are related to HIV (by concluding this in the abstract and discussion) but based on the evidence provided, it's unclear why the alternative explanation (end of apartheid and increased travel) is not equally likely. Also, given Shigella is endemic in settings without HIV, such as in South Asia, it's unclear how much weight should be given to the HIV argument, particularly how public health approaches that take HIV into account would benefit Shigella specifically. Can the authors elaborate.

391-392: Refers to the standard case definition however this is not easily ascertained through the references provided. What was the case definition of shigella suspects? This will help with interpreting the results.

395-396: Can the authors elaborate more on the missing data statement? Does this mean that 2014/2015 had too much missing data? This is helpful to understand generalizability.

Response to reviewers

General:

We thank the reviewers for taking the time to read our paper and provide feedback, which has led to its improvement.

Reviewer #1 (Remarks to the Author):

The authors present a very valuable study into the genomic epidemiology of *Shigella* in south Africa. They report new insights, especially regarding the difference in epidemiology of the different species. Technically the study is sound. Few comments that might improve the paper:

1. The authors talk about "serotype" specific epidemiology; however, to my knowledge the authors compare two different *Shigella* species (*flexneri* and *sonnei*) rather than two serotypes. Throughout the manuscript this is creating confusing.

We appreciate the reviewer's point, but in this case, the comparison is between *Shigella sonnei*, of which there is only one serotype, and *Shigella flexneri* serotype 2a so the use of the word "serotypes" throughout is more accurate than "species".

2. In several parts of the manuscript the authors talk about "ecology" (ecology as a driver of shigellosis epidemiology). It is unclear what the authors exactly mean by this and needs further explanation.

Thanks for raising this. In this case we refer to the lifestyle dynamics of the pathogen in regard to endemicity and potential reservoirs vs. importation, etc. To address the reviewer's comment, we have changed the "ecology" to the more specific "lifestyle" in the abstract (Line 26) as this is the most strongly supported link with driving epidemiological differences between *S. sonnei* and *S. flexneri*. In the discussion we have added a sentence linking pathogen ecology and lifestyle (Line 328). In the concluding paragraph (Line 368) We have also changed ecology to lifestyle.

3. It is not clear to what extent patient information is available. It would be high beneficial for to know the difference in incidence of both species in MSM versus non-MSM. it might also help to explain the difference in population ecology, and why amr is not necessary for *flexneri* success (probably because it finds its niche in non-msm more with lower antimicrobial use?)

No sexuality data was collected from the patient, nor did we find any indicators of MSM-associated lineages when looking at the distribution of age and sex across the phylogenies (specifically, there were no lineages enriched for male patients which has been used effectively in other studies and in public health surveillance). We believe this is due to the high prevalence of endemic shigellosis in the general population obscuring the signal, rather than necessarily an absence of MSM-associated transmission. To make this more obvious to readers we have added a sentence (Line 413) in the methods to indicate the lack of sexuality data and another in the discussion section (Line 303) to highlight the lack of MSM-associated signal in the phylogenies.

4. Extended data figure 3: why is for *sonnei* possible to separate between different age-groups and gender, and not for *flexneri*?

We thank the reviewer for this feedback which has arisen from a lack of clarity in the Figure. The HIV data is not presented with reference to the serotype headings at the top of the Figure (which relate only to parts A and B). This is clear from the Figure legend, but we have made a modification to the Figure also to make this clearer.

Reviewer #2 (Remarks to the Author):

GENERAL: This is a nicely presented article on the genomic epidemiology of Shigella infections in South Africa. The manuscript was very clear and easy to read, although there were some typographical errors throughout. The genomic comparison between Sh. sonnei and Sh. flexneri in the South African context was very interesting. I thought that the BEAST analysis was good showing timescales for potential introduction of strains into South Africa. The data are novel in that they come from surveillance data over a longer period of time than other studies conducted in Africa, such as the GEMS study.

Overall, the methodology was sound, based on genomic analysis of isolates collected through a surveillance program, although I did think that the statistical analysis of the geographic and temporal trends did not take into account the way the data were collected. What were the potential biases introduced into the data through the collection methods of the surveillance? It is disappointing that there wasn't good epidemiological data associated with the study isolate. This led to assumptions and speculation in some places in the manuscript, which were never acknowledged as a limitation.

We agree with the reviewer's assessment of the limitations of a surveillance-based dataset, particularly without surveillance coverage data to assess the introduced bias. However, we feel this represents a particularly valuable study that has approximated the genuine picture in an endemic African country. To acknowledge the potential for bias, we have explicitly discussed this as a limitation of the study in the discussion. See Line 361: "Another limitation of this study relates to the paucity of surveillance data for gastrointestinal illnesses in endemic regions. We have tried to mitigate this by carefully selecting our subsample relative to case reporting data, but the potential for sampling bias to have impacted our overall findings cannot be completely ruled out."

SPECIFIC:

L107: Doesn't this just indicate that there may have been outbreaks of these strains in these provinces in these time periods? It doesn't really go to the problem of sampling bias from the surveillance system itself, which is the real problem that is difficult to account for.

Thanks for raising this point, we have changed the term "sampling bias" to "sub-sampling bias" (Line 109) as this paragraph was meant to show our data was representative of the larger surveillance dataset, not assess sampling bias inherent in the surveillance dataset. Minimal data was available regarding surveillance coverage for us to assess the likely biases introduced through imperfect surveillance, though we acknowledge that the surveillance was indeed imperfect. To address the reviewer's concern (and in line with the above) we have explicitly acknowledged the limitation of sampling bias in the discussion (see above) and reworded the scope of our findings to "across the study period" to Line 108.

L129: Just to note that you haven't presented average ages, but medians and IQRs.

We thank the reviewer for highlighting this minor error and have changed the "average" to "median" in Line 131.

L136: I don't think that the reference to the Orthodox Jewish community is necessary given that we have always known that caring for a patient with shigellosis is a risk factor for infection.

We agree with the reviewer and have removed explicit mention of the Orthodox Jewish community (Line 138).

Line 182: Note missing reference

Line 245: Note missing reference

Thanks for pointing this out, we have re-added the missing extended data figure reference (Line 247) and removed the erroneous "missing reference" (Line 185).

Line 290: Wording is clumsy ('...supported by estimated population increase population size...'). I am not sure what is meant here.

Thanks for highlighting this, we have made this line (Line 294) more readable.

Line 307: Strains is spelt incorrectly 'stains'.

We have corrected the spelling (Line 312).

Line 309: The comparison of South Africa with South East Asia was confusing as South East Asia isn't the same geographical size as South Africa. Were you meaning to compare your study findings with that of Chung et al (ref #17)? I might have missed something here.

Yes, this was a comparison to the cited study. We have edited Line 313 to make this more obvious.

L325: I think it is speculation to say that 'epidemiological success in MSM is perhaps linked to AMR...'. I don't think we can really know the reason for persistence of these MDR strains in MSM from genomics studies alone.

We confirm that it is a somewhat speculative statement for the reasons stated, however, we also feel there is enough data supporting AMR promoting epidemiological success in bacteria generally, in *Shigella* and also within the MSM community specifically to justify its inclusion when contextualising our results. To make the speculation clearer to the reader we have modified the statement (Lines 331-333).

L331: I don't think you can claim that there hasn't been introduction of less drug susceptible *Shigella* strain, as countries see persistence of strains with different characteristics all the time, they just don't publish it.

We have changed the wording of this to reported (Line 336).

L364: What is the total number of biochemically identified *Shigella* and *Sh. flexneri* and *sonnei*? It is important for readers to understand what proportion your sample makes up of the population of isolates.

We have added the relevant *Shigella* isolate totals for surveillance during the study period at Line 381.

L366: It would help the reader to briefly explain the GERMS-SA surveillance and how isolates were collected and from whom. This could be done in more detail in L389. I just don't get a feel for who gets tested for Shigella and when. What kind of undercount was there in surveillance data?

We added further detail and some detail on the limitations at Lines 361-366 to provide more context for the surveillance carried out. Further information can also be found in the Data collection section (starting Line 405). It is unclear how much undercount there is from the annual reports, though Figure 1 provides greater context for the provinces sampled from for this study.

L390: It is important to discuss in the manuscript that the observed differences relate to the nature and performance of surveillance.

To highlight the limitations of the study due to imperfect surveillance we have extended a paragraph explicitly highlighting the limitations introduced in imperfect surveillance (Lines 365 and 366). However, importantly, any systematic bias in sampling would have been equal for *S. sonnei* and *S. flexneri* 2a so there is no reason why these head to head comparisons should not be valid.

L510: Where does the national total come from? I really didn't get a feel for national rates of shigellosis in South Africa either.

The national data comes from the GERMS-SA annual reports, this is referred to in the Data collection section of the methods (Lines 404 to 407), though to make this clearer in the statistical analysis section that this is where the numbers come from, we have added this to Line 517. The national incidence estimate is also included in the introduction.

L514: You mention using odds ratios although I couldn't see them anywhere in the paper. If you have used them, it would be better to present them with 95% confidence intervals than p values. I am not convinced of the need of Bonferroni correction.

Odds ratios were used for comparisons shown in the supplementary data, none are reported directly in the article, where they are reported with both 95% CI and p-values. We used the Bonferroni correction as there were around 16 individual comparisons for one of the serotypes.

Reviewer #3 (Remarks to the Author):

This is an excellent manuscript describing the genomic epidemiology of Shigella in South Africa, marking an important contribution to the dearth of evidence around Shigella epidemiology in the sub-Saharan Africa continent. Most impressive in my mind, is this enteric surveillance system that is set up in South Africa providing an incredibly rich isolate and data repository enabling this type of analysis.

That age of infection was older in females than in males is interesting, and to the best of my knowledge, this has not been demonstrated previously. In addition to caregiving responsibilities, could this also be explained by the surveillance system and/or culturability of the isolates? For example, it's been shown that Shigella isolate recovery is more common in older children (ie culture negative/ PCR positive infection more common in young children). Shigella can be tricky to recover, and is impacted by transport media etc. Could it also be that surveillance facilities with more female participants had a longer transport time to the lab?

Thanks for the feedback. While it is possible that culture rates differ between the sexes, we know of no study showing this and we don't feel we have enough evidence or a speculative mechanism to support the inclusion of this hypothesis in the article, given the evidence that already exists for the role of care giving. We also have no evidence of sex-based differences in hospital attendance in South Africa, though it is possible that adult women could be more likely to be sampled from than adult men due to greater health seeking behaviours in women.

The authors conclude that the new Shigella strains in the 1990s are related to HIV (by concluding this in the abstract and discussion) but based on the evidence provided, it's unclear why the alternative explanation (end of apartheid and increased travel) is not equally likely. Also, given Shigella is endemic in settings without HIV, such as in South Asia, it's unclear how much weight should be given to the HIV argument, particularly how public health approaches that take HIV into account would benefit Shigella specifically. Can the authors elaborate.

Thanks for raising this point. We feel that both are likely influencing factors and that our study is unable to point to one or the other as more important. The focus more on HIV as a factor was driven by prior evidence of HIV being a risk factor for shigellosis (from sexual transmission studies), and because the impact of HIV is an easier target for public healthcare surveillance and intervention. To reflect this reasoning, while ensuring that the role of the ending of Apartheid is not understated, we have added a sentence at Lines 358 and 359. And edited the following sentence (Line 360).

391-392: Refers to the standard case definition however this is not easily ascertained through the references provided. What was the case definition of shigella suspects? This will help with interpreting the results.

We have added details for the surveillance procedures (Lines 408-413).

395-396: Can the authors elaborate more on the missing data statement? Does this mean that 2014/2015 had too much missing data? This is helpful to understand generalizability.

The missing data was from the 2014 and 2015 GERMS-SA reports, not from our study data. We have made this clearer in Line 408.